# 360° Ab-Interno Schlemm’s Canal Viscodilation with OMNI Viscosurgical Systems for Open-Angle Glaucoma—Midterm Results

**DOI:** 10.3390/jcm11010259

**Published:** 2022-01-04

**Authors:** Giacomo Toneatto, Marco Zeppieri, Veronica Papa, Laura Rizzi, Carlo Salati, Andrea Gabai, Paolo Brusini

**Affiliations:** 1Department of Ophthalmology, University Hospital of Udine, 33100 Udine, Italy; giacomo.toneatto@gmail.com (G.T.); carlo.salati66@gmail.com (C.S.); andrea.gabai@gmail.com (A.G.); 2Department of Ophthalmology, Policlinico “Città di Udine”, 33100 Udine, Italy; papa.veronica87@gmail.com (V.P.); brusini@libero.it (P.B.); 3Department of Economics and Statistics, University of Udine, 33100 Udine, Italy; laura.rizzi@uniud.it

**Keywords:** open angle glaucoma (OAG), Schlemm’s canal viscodilation, OMNI viscosurgical system, minimally invasive glaucoma surgeries (MIGS), trabeculotomy, cataract extraction

## Abstract

Purpose: To evaluate the effectiveness of ab-interno microcatheterization and 360° viscodilation of Schlemm’s canal (SC) performed with OMNI viscosurgical system in open angle glaucoma (OAG) together or not with phacoemulsification. Setting: Two surgical sites. Design: Retrospective, observational. Methods: Eighty eyes from 73 patients with mild to moderate OAG underwent ab- interno SC viscodilation performed with OMNI system. Fifty eyes (Group 1) underwent only SC viscodilation, while 30 eyes (Group 2) underwent glaucoma surgery + cataract extraction. Primary success endpoint at 12 months was an intraocular pressure (IOP) reduction higher than 25% from baseline with an absolute value of 18 mmHg or lower, either on the same number or fewer ocular hypotensive medications, without further interventions. Secondary effectiveness endpoints included mean IOP, number of medications and comparison of outcomes between groups. Safety endpoints consisted of best-corrected visual acuity (BCVA), adverse events (AEs), and subsequent surgical procedures. Results: Primary success was achieved in 40.0% and 67.9% in Groups 1 and 2, respectively. Mean IOP at 12-month follow-up showed a significant reduction in both groups (from 23.0 to 15.6 mmHg, *p* < 0.001, and from 21.5 to 14.1, *p* < 0.001, in Groups 1 and 2, respectively). Mean medication number decreased in both groups (from 3.0 to 2.0, *p* < 0.001 and from 3.4 to 1.9, *p* < 0.001, in Groups 1 and 2, respectively). AEs included hyphema (2 eyes), mild hypotony (4 eyes), IOP spikes one month after surgery (1 eye). Twelve eyes (15.0%) required subsequent surgical procedures. No BCVA reduction was observed. Conclusions: Viscodilation of SC using OMNI viscosurgical systems is safe and relatively effective in reducing IOP in adult patients with OAG.

## 1. Introduction

Glaucoma is one of the leading worldwide causes of blindness [1]. The most important risk factor for development and progression of glaucoma is elevated intraocular pressure (IOP) [2]. Important clinical trials such as the Ocular Hypertension Treatment Study (OHTS) and Advanced Glaucoma Intervention Study (AGIS) established the importance of IOP reduction in glaucoma management [3,4]. The first therapeutic option is the use of ocular hypotensive eye drops; however, medical treatment tends to be associated with poor compliance and tolerability [5,6]. When medical treatment proves to be insufficient to reach the target IOP or drops are not well tolerated, laser or surgical treatment need to be considered to avoid irreversible damage progression.

Trabeculectomy, usually with antimetabolites, is still considered the gold standard surgical procedure [7], however, it is not free of potentially serious complications [8,9]. In addition, strict postoperative care is mandatory to obtain clinically successful results. Minimally invasive glaucoma surgeries (MIGS) have been developed as safer and less invasive techniques. MIGS are characterized by an ab-interno approach, minimal trauma and disruption of eye anatomy with conjunctiva sparing, high safety profile, rapid recovery and the possibility of performing the procedure during routine cataract surgery [10].

Ab-interno Schlemm’s canal (SC) viscodilation, performed with the OMNI device is a novel MIGS angle surgical procedure. The aim of this study was to evaluate the effectiveness of ab-interno microcatheterization and 360° viscodilation of SC performed with the OMNI viscosurgical system in adult patients with open angle glaucoma (OAG).

## 2. Materials and Methods

The investigation was based on a double-center, retrospective, observational, consecutive design. All patients underwent surgery with the OMNI viscosurgical system (Sight Sciences Inc., Menlo Park, CA, USA) by 3 ophthalmic surgeons (G.T., C.S., P.B.) from 2 multi-subspecialty ophthalmic departments (University Hospital of Udine and Policlinico “Città di Udine”). All surgeries were performed between March 2017 and January 2020. All adult patients (>18 years old) with mild to moderate (Glaucoma Staging System 2 Stages 1 to 3) [10]. OAG who had undergone ab-interno SC viscodilation were enrolled retrospectively. The types of OAGs considered included primary open angle glaucoma (POAG), pigmentary glaucoma (PG) and pseudoexfoliative glaucoma (XFG). Patients underwent surgery for uncontrolled IOP values on maximally tolerated therapy or to reduce medical therapy due compliance and/or tolerance issues. The ab-interno SC viscodilation surgical procedure was combined with cataract extraction when indicated.

### 2.1. Main Outcome Measures

Each patient underwent a complete baseline ophthalmologic examination, including ocular history, ophthalmic and systemic medication used, best corrected visual acuity (BCVA) using Snellen charts, IOP measured by Goldmann applanation tonometry, central corneal thickness, gonioscopy, undilated and dilated slit-lamp biomicroscopic examination. Cup disc ratio (CDR), measured by slit-lamp fundus biomicroscopy in relation with disc size, visual field examination using 24-2 SITA Standard test (Humphrey Field Analyzer II; Carl Zeiss Meditec Inc., Dublin, CA, USA), and measurement of macular ganglion cell and retinal nerve fiber layer thickness using optical coherence tomography (OCT RS-3000 Advance; Nidek CO. LTD., 34-14 Maehama, Hiroishi-cho, Gamagori, Aichi 443-0038, JAPAN and DRI-OCT Triton, Topcon, Inc, Tokyo, Japan) were used to evaluate the severity of glaucomatous damage and to determine target pressure before planning glaucoma surgery. Follow-up examinations were performed at Week 1, Week 2, Month 1, Month 3, Month 6, and Month 12. At each follow-up, information regarding IOP, number of antiglaucoma medications, BCVA, postoperative AEs and any other interventions was recorded and used in the analysis.

The study cohort was divided into two groups based on whether or not combined cataract surgery was performed. Group 1 included all eyes that underwent ab interno microcatheterization and viscodilation of SC as a standalone procedure, while Group 2 included all eyes with glaucoma surgery combined with cataract extraction. Our primary success endpoint at the 12-month follow-up visit was defined as a reduction of IOP equal to or greater than 25% from baseline with an absolute value of 18 mmHg or lower, either with the same number of ocular hypotensive medications or fewer, with no additional IOP-lowering surgery. The eyes who reached this endpoint without any medical treatment were considered as complete success, whereas those where this result was obtained using medical treatment were labelled as qualified success. The secondary endpoints included mean IOP and mean number of ocular hypotensive medications at each follow-up time up to 12 months. Outcomes were also compared between groups.

The investigation was performed in accordance with the tenets of the Declaration of Helsinki, and informed consent was obtained from all participants before surgery. The study was in compliance with institutional review boards (IRBs) and HIPAA requirements of the University Hospital of Udine, Italy.

### 2.2. Statistical Methods

The mean levels of IOP at different follow up times were compared to baseline by means of paired sample *t*-test, which was also used to compare IOP mean reduction at 12 months in Groups 1 and 2. The decrease of medications in each group was analyzed by means of non-parametric Wilcoxon matched-pairs signed rank test. The comparison between Group 1 and 2 on the ordinal variable of the difference between medications at 12 months and at pre-surgery time was performed using Mann–Whitney U test. Descriptive and survival analysis was used to describe proportions of success, which was defined in terms of specific levels of IOP target reached with or without antiglaucoma medications, and to compare failure probabilities in the two groups. BCVA changes at follow-up times were evaluated and compared at group level with parametric *t*-test for paired data.

### 2.3. Surgical Technique

The surgical technique for ab-interno SC viscodilation as a sole procedure includes the following steps.

After a 1.5 mm paracentesis, pupil miosis was obtained by injecting acetylcholine chloride 1% (Miovisin; Farmigea SPA, Pisa, Italy) into the anterior chamber. Space and stability were achieved by cohesive ophthalmic viscosurgical device (Healon GV; Johnson & Johnson Surgical Vision, Santa Ana, CA, USA). A moderate intraocular pressure was maintained in order to enhance visualization of SC during surgery. The head was tilted and positioned about 40° from the surgeon, while the microscope was tilted 40° towards the surgeon for gonioscopic visualization of the trabecular meshwork.

The OMNI Viscosurgical System (Figure 1a shows the detail of the cannula and the microcatheter) was prepared by removing the retainer pin on the back of the handle and by loading the reservoir with Healon GV (Figure 1b). The cannula was then introduced into the anterior chamber by a 1.5 mm, self-sealing, clear corneal incision performed in the temporal region. The cannula tip was brought near the nasal trabecular meshwork under gonioscopic view (ocular surgical gonioprism for direct viewing gonioscopy, Figure 1c) and used to gently but firmly open the SC. The microcatheter was then advanced into SC for 180° by rotating a finger wheel on the handle of the device (Figure 1d). As the microcatheter was retracted into the cannula, the infusion pump delivered a controlled volume of Healon GV to achieve viscodilation of SC and collectors. The same procedure was then repeated to expand the other 180° of SC.

The surgeon had to be careful to pinch the trabecular meshwork in the right location and to clearly see the blue microcannula into the SC to avoid undesirable suprachoroidal advancement of the tip. Blood reflux indicates a successful catheterization of SC. As the cannula was retracted out the eye, the anterior chamber was irrigated to entirely remove Healon GV.

With regards to ab-interno SC viscodilation combined with cataract surgery, the surgical steps are slightly modified. A 2.2 mm microincision cataract surgery (MICS) was performed with the Constellation Vision System (Alcon Laboratories, Fort Worth, TX, USA). All IOLs were implanted in the capsular bag, which were all hydrophobic, acrylic and monofocal. Acetylcholine chloride 1% (Miovisin; Farmigea SPA, Pisa, Italy) was then injected into the anterior chamber to obtain pupil miosis. Viscodilation of SC was achieved with the method previously described.

## 3. Results

The study cohort consisted of 80 eyes of 73 patients that met the inclusion criteria and completed the baseline evaluation, with 66 eyes (82%) completing the 12-month visit. Of the remaining 14 eyes, 12 (15%) had additional glaucoma surgery and 2 (2.5%) were lost to follow-up. The patients were divided into two groups according to the type of surgery performed: Group 1 (SC viscodilation alone) consisted of 50 eyes (62.5%) while Group 2 (SC viscodilation combined with phaco) included 30 eyes (37.5%). Table 1 shows the baseline characteristics. The patients were predominantly female, with the majority diagnosed with POAG (63.8%). The most common previous anti-glaucoma procedure was laser trabeculoplasty (19%). Thirteen patients (16.2%) had previous glaucoma surgery (10 cases underwent deep sclerectomy, and three cases had trabeculectomy) that did not interfere with the complete catheterization of Schlemm’s canal. Forty-four eyes (55%) were pseudophakic at baseline. Of the remaining 36 eyes, 30 of them (83%) had combined cataract extraction with IOL implantation. Successful circumferential catheterization of the canal was achieved in all eyes and no significant adverse events were recorded intraoperatively, other than small bleeding from the SC.

### 3.1. Differences in Intraocular Pressure and Antiglaucoma Medication Used

Table 2 shows the differences in intraocular pressure and antiglaucoma medications used at the different time points overall and in the two groups. Mean IOP values at baseline were 22.5 ± 5.3 mmHg for all eyes, and 23.0 ± 5.7 mmHg and 21.5 ± 4.7 mmHg in Group 1 and 2, respectively. After 12 months, mean IOP reduced to 15.0 ± 3.6 mmHg, 15.6 ± 3.6 mmHg, and 14.1 ± 3.3 mmHg, respectively. IOP reduction was statistically significant in both groups at 1, 3, 6 and 12 months (*p* < 0.001). The difference between mean IOP reduction in Groups 1 and 2, however, was not statistically significant at 12 months (*p* = 0.21). Figure 2A reports the box plots of IOP distributions at different follow up times in all the patients and in the two groups.

The mean number of medications at baseline was 3.0 ± 1.1 and 3.4 ± 0.8 in Group 1 and 2, respectively, and decreased at 12 months to 2.0 ± 1.4 and 1.9 ± 1.4, respectively (Figure 2B). Wilcoxon matched pairs signed rank test showed statistically significant reductions in all the groups and at all time points (*p* < 0.001). The reduction of medications at 12 months between the two groups was not significantly different (*p* = 0.26).

### 3.2. Success

Table 3 reports the proportions of complete and qualified success with different levels of IOP target at 12 months over the entire cohort and for Groups 1 and 2. The success was defined as complete if a specific IOP level was reached without antiglaucoma medications, and as qualified, if target IOP was achieved using medications. The IOP endpoints chosen to stratify success results were <16, <18 and <21 mmHg. Moreover, outcomes were more strictly sorted by a 25% reduction in IOP from the baseline value. At 12 months of follow-up, 14.0% of Group 1 eyes reached an IOP of 18 mmHg or lower with no medications and 66.0% achieved success either by the use or without medications. In Group 2, 17.9% of eyes reached an IOP of 18 mmHg or lower with surgery alone and 82.1% achieved a qualified success. Overall, 15.4% of eyes attained an IOP of 18 mmHg or lower with no medications and 71.8% achieved a qualified success.

Considering an additional reduction of IOP higher than 25% to estimate our primary success endpoint, 40.0% and 67.9% of eyes reached an IOP of 18 mmHg or lower with or without medications in Group 1 and in Group 2, respectively.

Figure 3 shows the observations on the preoperative IOP values (horizontal axis) and the postoperative ones (vertical axis). The horizontal lines represent the different IOP endpoints (21, 18 and 16 mmHg), while diagonal lines define null and 25% reduction level of IOP from baseline.

Figure 4 reports the Kaplan–Meier survival plots for probability of qualified success in Group 1 and Group 2, using the failure criterion of an IOP value higher than 18 mmHg in three subsequent visits. Log-rank Chi-squared test suggested the acceptance of the hypothesis on the equality of survivor estimates in the two groups (*p* = 0.28).

### 3.3. Safety

After twelve months of follow up, no significant change in BCVA was observed in Group 1 (from 7.9/10 ± 2.4 to 7.7/10 ± 2.9, *p* = 0.31). A BCVA gain, after 12 months, resulted statistically significant only in Group 2 (from 6.8/10 ± 2.1 to 9.1/10 ± 2.6, *p* < 0.001), due to the combination of SC viscodilation with phaco.

Most complications were recorded in the first follow-up week. During the surgical procedure, a mild blood reflux in AC was considered a sign of successful catheterization of SC, occurring in almost 100% of our patients. Cases of micro or mild hyphema on the first postoperative day, defined respectively as circulating red blood cells without a blood level, and as blood layer filling less than 1 mm in the anterior chamber, were not considered as AEs, given the physiological and self-limiting nature of these occurrences. Two eyes (2.5%) developed moderate and severe hyphema and were reported as AEs. The first one showed a 3 mm layer of blood combined with peaks of high IOP 2 days after surgery. The second case reported a sustained hyphema filling more than half of the AC associated with BCVA reduction to hand motion without a rise of IOP. Both cases required a wash of the AC, three and ten days after surgery respectively, with complete recovery.

Four eyes (5.0%) had mild hypotony (4–5 mmHg) within the first month postoperatively. No shallow anterior chamber or choroidal detachment were detected. All cases resolved without any intervention. One eye (1.3%) reached an IOP level >10 mmHg above baseline more than 30 days postoperatively.

Twelve subjects (15.0%) required a secondary intervention due to an unreached IOP target level: trabeculectomy in nine cases (75%), and deep sclerectomy in three cases (25%). The mean IOP and number of medications before these surgical procedures were 26.3 ± 6.5 mmHg and 2.8 ± 1.3, respectively. One phakic eye needed cataract extraction 6 months after surgery. Three eyes developed complications unrelated to glaucoma surgery within 6 and 12 months: hypertensive anterior uveitis successfully treated with topical steroid (one eye), active choroidal neovascularization due to age-related macular degeneration (one eye), posterior capsular opacity needing Nd:YAG laser capsulotomy (one eye).

## 4. Discussion

Trabeculectomy has been the gold standard in glaucoma surgery since 1968 [11]. Although this procedure continues to have a significant role due to the capacity to achieve long-standing significant IOP lowering, non-penetrating filtering surgeries (NPFS) have been proposed to achieve IOP reduction avoiding serious sight threatening complications such as shallow anterior chamber, uncontrolled hypotension, choroidal detachment and macular folds. Outflow resistance can be attributed to three structures: the juxtacanalicular meshwork, SC and collector channels [12]. Non penetrating filtering surgery (NPFS) such as canoloplasty, include techniques focused on the dilation of SC to facilitate aqueous outflow through the physiological route. The aim is to remove mechanical obstructions in the collector channels, by enhancing and providing additional routes for aqueous outflow. Viscodilation separates the trabecular lamellae and creates microperforations within the inner wall of SC, allowing for enhanced diffusion of aqueous through the proximal system into the distal system and thereby countering the pathological changes seen in glaucoma [13].

Preliminary procedures focusing on SC were historically sinusotomy described by Kraznov in 1962, followed by Stegmann’s viscocanalostomy [14,15]. Ab externo canaloplasty (ABeC), proposed by Lewis and coworkers in 2007, is a surgical technique that uses a microcatheter to perform a 360° cannulation of SC positioning a tension suture within the canal that provides an inward distension [16]. The aim of this technique is to restore the physiological outflow pathways of the aqueous humor independently of external wound healing. Numerous studies have shown that canaloplasty is a relatively safe and effective surgical technique that lowers IOP with persistent control of pressure during many years of follow-up. Moreover, it implies easier postoperative management and less complications compared with trabeculectomy [17,18,19].

In the past several years, there has been a gain in popularity of MIGS to address the need of achieving long-term IOP control in the safest possible way. MIGS is characterized by an ab-interno approach, inducing minimal trauma and disruption of eye anatomy with conjunctiva sparing and a rapid recovery [10]. With the advent of MIGS, surgeons have been choosing to perform earlier surgery in patients with mild to moderate glaucoma in order to achieve the target IOP with fewer medications [20].

A minimally invasive technique, called ab-interno canaloplasty (ABiC), has recently been developed to reap the advantages of ABeC, avoiding conjunctival and scleral dissection [21]. Considering that ABeC and ABiC tend to have a rather long and steep learning curve, OMNI viscosurgical system has been refined to allow an easier cannulation of SC and a more standardized viscodilation of outflow pathways. Moreover, ABeC is not always successful in providing a 360° catheterization of SC and failure rates range from 10.1 to 26% [22], which can be due to anatomical anomalies of SC, trabecular meshwork scars, neovascularization of iridocorneal angle, but also lack of surgical experience. In our study, 100% of complete catheterization of SC was achieved due to a good preliminary gonioscopic evaluation of angle structures and to “surgeon friendly” characteristics of the device.

Both ABeC and ABiC utilize an illuminated microcatheter to access, catheterize, and viscodilate the proximal and distal outflow system. The OMNI viscosurgical system is a single-handed device, equipped with a metallic cannula that encases a microcatheter, control wheel for advancing and retracting microcatheter, viscoelastic reservoir/infusion pump and a locking mechanism. These devices facilitate automatic delivery of a predetermined amount of viscoelastic fluid to dilate 360° of SC. Although OMNI has been designed to perform, when desired, a secondary trabeculotomy in addition to viscodilation of SC to treat juxtacanalicular meshwork resistances, we used the device only to viscodilate without unroofing the SC.

Our primary success endpoint at 12-month follow-up visit was defined as the proportion of eyes achieving an IOP value equal or under 18 mmHg with a reduction higher than 25% from baseline, either on the same number or fewer ocular hypotensive medications, and with no additional IOP-lowering surgery or laser. This choice was based on the baseline characteristics of our cohort of patients, including only mild to moderate open angle glaucomas, in which intraocular pressure of 18 mmHg or below was an acceptable criterion for controlling progression. The proportions of success defined with the endpoints of 16 and 21 mmHg, with an IOP reduction higher than 25% from baseline, were also analyzed.

In our study, 40% and 67.9% of eyes reached the primary endpoint at 12 months in Group 1 and Group 2 respectively. These results appeared to be worse than the outcomes of ABeC as a standalone procedure (68.1%) and combined with phaco (77.8%) [17], probably because of the presence of the suture that provides a more durable distension of SC.

In our cohort, the mean IOP reduction at 12 months compared to baseline was of 26.8% in Group 1 (from 23.0 ± 5.7 mmHg to 15.6 ± 3.6 mmHg) and 32.4% in Group 2 (from 21.5 ± 4.7 to 14.1 ± 3.3). The IOP reduction was statistically significant in both groups during the entire follow-up. The difference in IOP reduction at 12 months in Group 2, compared with Group 1, was not statistically significant. The overall reduction in IOP in the entire cohort was 29.0% at 12 months (from 22.5 ± 5.3 mmHg to 15.0 ± 3.6 mmHg). Similar IOP changes have been described for ABeC [17,18] and ABiC [21,23,24,25] in previous studies. Brusini [19] reported a more favorable reduction in IOP from 29.4 ± 7.9 preoperatively to 16.8 ± 4.2 mmHg at 12 months probably due to the higher IOP level at baseline compared with the other studies.

Our surgical goal was decreasing IOP and/or reducing glaucoma drops. The mean number of medications at 12 months decreased from 3.0 ± 1.1 to 2.0 ± 1.4, in Group 1, and from 3.4 ± 0.8 to 1.9 ± 1.4, in Group 2, and was statistically significant in all groups at all time points. Compared with other clinical studies, our patients tended to be on a higher number of medications before surgery. The mean reduction of antiglaucoma drops of 1.0 and 1.6 respectively for Group 1 and Group 2 after 12 months was similar to ABeC and ABiC [17,18,23].

The results showed that surgery provided effective IOP control and reduction, with few AEs. Self-limiting microhyphema was frequently observed due to the physiological blood reflux from SC, thus not considered as an AE. Only clinically significant hyphemas (i.e., layered and >1 mm and persisting for 1 week or more) were recorded as AEs in this study. One eye reported a 3 mm hyphema associated with an important postoperative IOP elevation, while another case presented with a blood level filling more than half of AC associated with an important decrease of visual acuity. These two eyes (2.5%) underwent a washout of the anterior chamber three and ten days after surgery, respectively, without further postoperative sight threatening complications. These AEs occurred even if the surgeons carefully screened for and avoided patients from taking anticoagulant therapy during the perioperative period, whenever possible, and were careful in maintaining an adequate pressure of the globe at the end of the procedure. Ondrejka et al. [26] reported a 2.8% rate of hyphema between 1 and 3 mm that spontaneously resolved within 7 days, which was similar to the 2.5% rate reported in our results. Vold et al. [27] reported a slightly greater incidence of postoperative clinically significant hyphema >1 mm (4%), in addition to Sarkisian et al. [28] (4.9% of moderate and severe hyphema) and Grabska-Liberek et al. [29] (35%, of which half required AC washout on the first postoperative day due to marked IOP elevation). The higher rate of hyphema could be due to the fact that the surgeons used the OMNI system to perform canaloplasty combined with trabeculotomy. The rates of hyphema for the ABiC procedure reported in literature ranges from 0% to 20% [21,23,24,25]. The higher rate of 20% was found in the study by Kazerounian [25], which refers to mild hyphema cases with no late sequelae.

In our study, four eyes developed mild postoperative hypotony (IOP 4–5 mmHg) within the first month but resolved without any intervention. No shallow anterior chamber or choroidal detachment were detected. IOP had major fluctuations within the first 30 days postoperative because of antiglaucoma drops discontinuation after surgery and the use of steroids during the first weeks. The incidence of IOP spikes was low after the first month of follow-up with only one eye (1.3%) reaching an IOP > 10 mmHg from baseline at 1 month. The use of the OMNI did not affect visual acuity.

The statistical analysis showed differences between groups at baseline. The mean preoperative BCVA in Group 2 was clearly lower compared to Group 1 due to the fact that Group 2 included only phakic eyes that were scheduled to perform cataract extraction combined to glaucoma surgery, in comparison with the predominately higher number of pseudophakic eyes in the other group. Patients in Group 2 showed a higher number of preoperative medications prior to surgery, however, the reduction of medications at 12 months between the two groups was not significantly different. In Group 1, a total of 10 and 3 eyes underwent previous deep sclerectomy and trabeculectomy respectively, while in Group 2, all eyes were naïve to prior surgery. The previous surgeries did not tend to interfere with the cannulation and viscodilation of SC using the OMNI system. With regards to deep sclerectomy, 4 of 10 eyes reached the primary endpoint at the 12-month follow-up visit, achieving an IOP value less than or equal to 18 mmHg and a reduction higher than 25% from baseline. This value was the same when compared with the results of the entire Group 1. This could probably be due to the fact that deep sclerectomy surgery tends to leave the trabecular meshwork undamaged, thus permitting a good dilation of SC and collectors by viscoelastic pressure during the OMNI system procedure. Due to the limited number of eyes in our cohort, it is difficult to establish the role of previous trabeculectomy when performing the OMNI procedure. Only one eye of three reached the primary success endpoint at 12 months.

Although there are a wide number of studies on ABeC and ABiC, literature is poor regarding outcomes of viscodilation of SC performed with OMNI system without trabeculotomy. Our results are similar to those reported for traditional ABeC and ABiC procedures. Viscodilation combined with trabeculotomy using OMNI viscosurgical system is considered an interesting option by some surgeons probably due to the possibility of treating the juxtacanalicular meshwork resistances as well as to enhance the outflow across SC and collectors [27,28,29,30]. Albeit with a similar effectiveness in IOP control and reduction, trabeculotomy seems to result in more AEs in terms of hyphema.

The OMNI system has several advantages compared to other MIGS procedures, which include: (1) unlike XENgel, it is not a filtering procedure, thus it is not mandatory that the conjunctiva is in good condition to obtain a filtering bleb; (2) it is easier to perform when compared to other similar techniques, such as ab-interno canaloplasty (ABiC); (3) it is more respectful of trabecular structures in comparison with trabectome or gonioscopy assisted transluminal trabeculotomy (GATT), which extensively open Schlemm’s canal and trabecular meshwork; and, (4) it could theoretically be more effective than i-Stent, however, studies have not been performed to date that compare these two techniques.

The OMNI system, however, has some disadvantages which include: (1) the difficulty to correctly find Schlemm’s canal in eyes with little or no pigment in the trabecular meshwork; (2) the impossibility to follow the microcatheter for the entire canal path, which can travel quite a distance far from the entrance; and (3) the possible bleeding into the anterior chamber, which seldom occurs.

The limitations of this study include the retrospective nature of collected data, the relatively low number of subjects included, the short follow-up period, and the enrollment of both eyes of the same patient in some cases. Furthermore, not all eyes were naïve to prior surgery: thirteen had already underwent previous major glaucoma surgery (trabeculectomy and deep sclerectomy). This choice was based on previous studies which have shown that canaloplasty can also be successfully performed in patients with failed trabeculectomy in which SC has been left mostly undamaged from previous filtering surgeries [31,32]. There was no standardized protocol for reducing or increasing medications, but the medical therapy was adjusted to reach the target IOP on a case-by-case basis. Follow-up was limited to 1 year. Due to the limited number of patients included in our cohort, it was difficult to assess the role and influence of glaucoma types on surgery between the groups. Future prospective studies are currently underway to help address this issue. Longer-term studies on a larger group of prospectively enrolled patients are needed to assess the duration of IOP reduction with this surgical technique.

In conclusion, viscodilation of SC using OMNI viscosurgical system, with or without cataract extraction, appears to be a promising surgical procedure to effectively control and reduce IOP with a highly safer profile, even if a high percentage of eyes require a medical treatment. Further studies are needed to report long-term results and complications, and to assess the real advantage of an associated trabeculotomy.

## Figures and Tables

**Figure 1 jcm-11-00259-f001:**
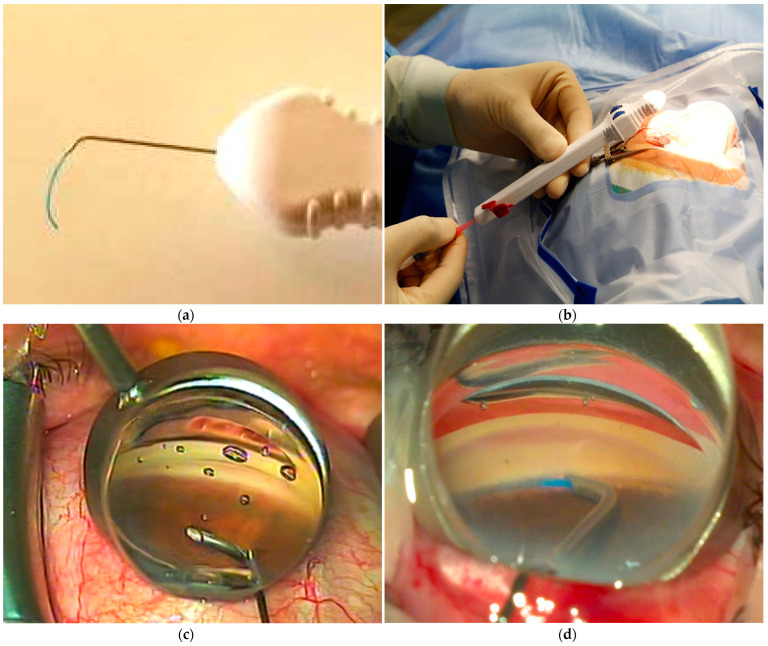
The OMNI device ends with a cannula and microcatheter at the tip (**a**). The preparation of the device includes loading the reservoir with high-molecular weight hyaluronic acid (**b**). The cannula is inserted into the anterior chamber under gonioscopic view to reach the nasal trabecular meshwork (**c**). The procedure involves the opening of trabecular meshwork and advancement of the microcatheter in Schlemm’s canal (**d**).

**Figure 2 jcm-11-00259-f002:**
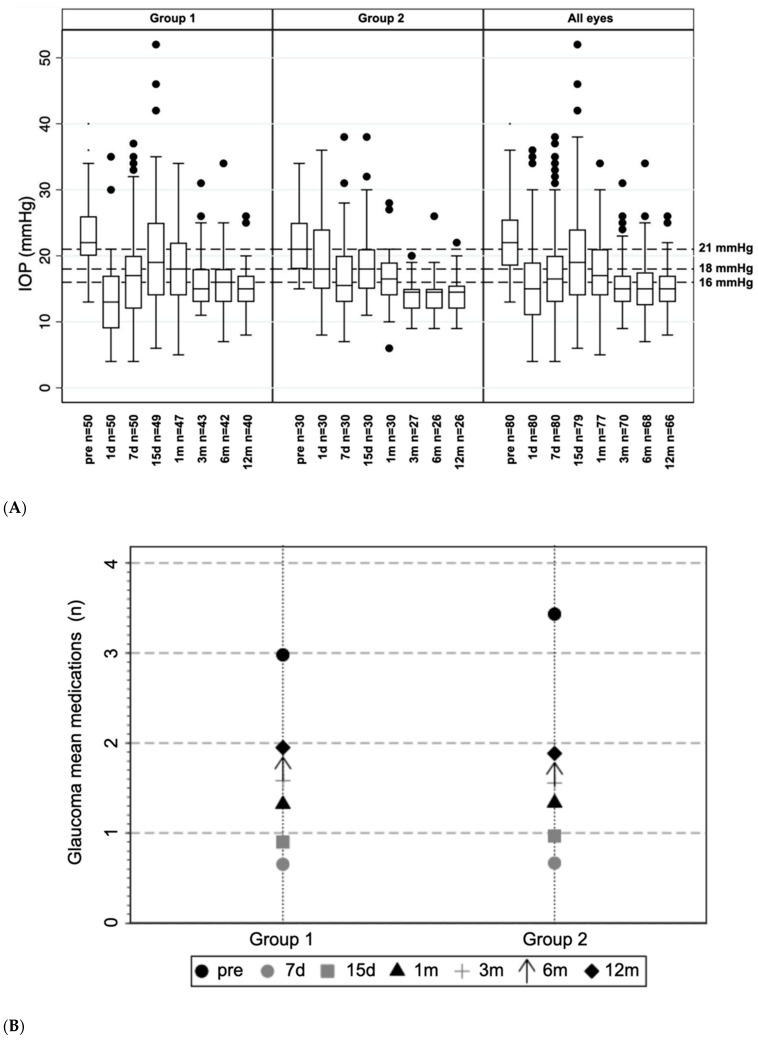
Boxplot of IOP distributions at baseline and at each follow-up visit for Groups 1 and 2 and for all the cohort ((**A**), above). Mean medications at the baseline and at each follow-up visit in Groups 1 and 2 ((**B**), below).

**Figure 3 jcm-11-00259-f003:**
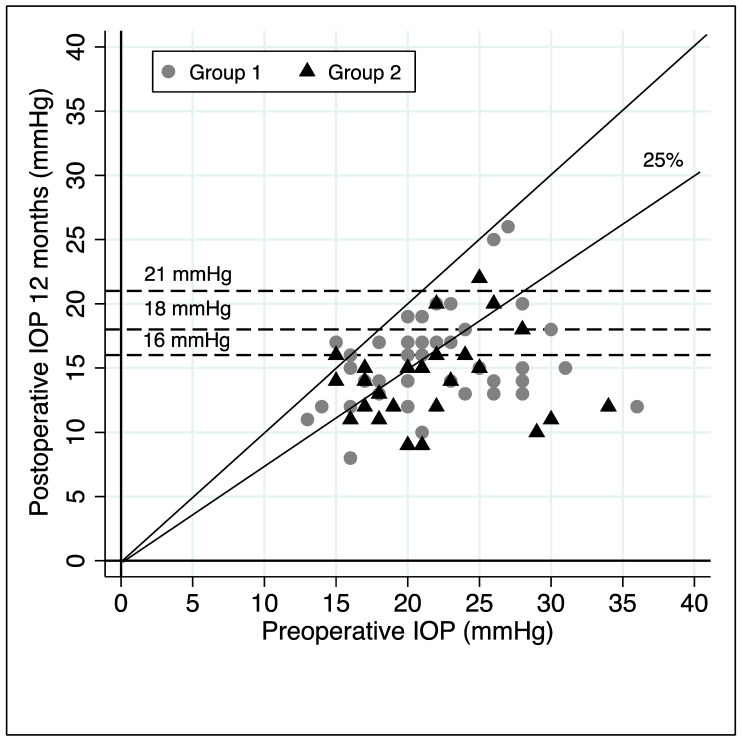
Scatter plot of observations on the pre- and 12-month postoperative IOP values, different IOP endpoints values (horizontal lines) and null or 25% reduction from baseline IOP diagonal lines.

**Figure 4 jcm-11-00259-f004:**
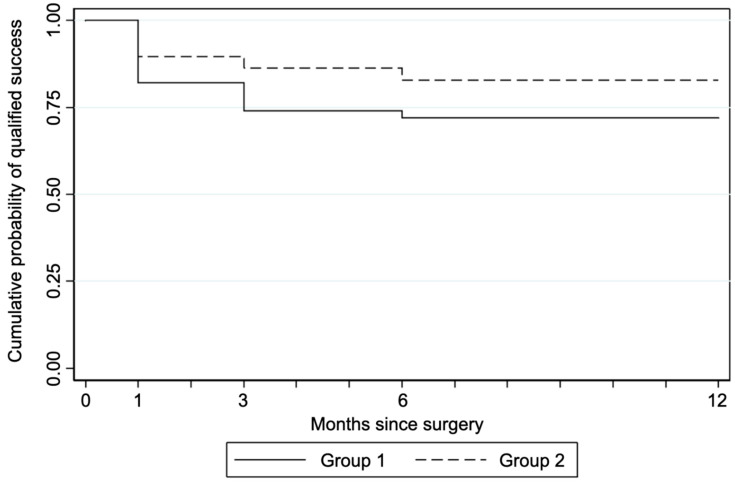
Kaplan–Meier survival estimates for Groups 1 and 2 (failure IOP > 18 mmHg at 3 consecutive follow up visits).

**Table 1 jcm-11-00259-t001:** Demographic and baseline characteristics.

	All Eyes	Group 1 (SC Viscodilation)	Group 2 (SC Vscd + Phaco)	Test on Differences (Statistical Significance)
**Eyes, *n***	80	50	30	
**Patients, *n***	73	47	26	
**Age in years**				
Mean Age ± SD (years)	74.5 ± 7.5	74.2 ± 8.0	75.0 ± 6.6	t test (*p* = 0.32)
Age Range (years)	56–93	56–93	62–89	
**Gender, *n* (%)**				
Female	43 (58.9)	26 (55.3)	17 (65.4)	Chi^2^ test on gender distribution (*p* = 0.35)
Male	30 (41.1)	21 (44.7)	9 (34.6)	
**Preoperative IOP, mean ± SD**	22.5 ± 5.3	23.0 ± 5.7	21.5 ± 4.7	t test (*p* = 0.11)
**Preoperative medications, mean n ± SD**	3.2 ± 1.0	3.0 ± 1.1	3.4 ± 0.8	t test (*p* = 0.03)
**Preoperative BCVA, mean ± SD**	7.5 ± 2.5	7.9 ± 2.4	6.8 ± 2.1	t test (*p* = 0.02)
**Glaucoma diagnosis, *n* (%)**				
Primary open angle glaucoma	51 (63.8)	37 (74.0)	14 (46.7)	Chi^2^ test (*p* = 0.01)
Pseudoexfoliative glaucoma	27 (33.7)	12 (24.0)	15 (50.0)	Chi^2^ test (*p* = 0.01)
Pigmentary dispersion glaucoma	2 (2.5)	1 (2.0)	1 (3.3)	Chi^2^ test (*p* = 0.71)
**Baseline lens status, *n* (%)**				
Pseudophakic	44 (55.0)	44 (88.0)	0 (0)	Chi^2^ test (*p* = 0.00)
Phakic	36 (45.0)	6 (12.0)	30 (100)	Chi^2^ test (*p* = 0.00)
**Previous surgery, *n* (%)**	-	-	-	
Cataract	44 (55.0)	44 (88.0)	0 (0.0)	Chi^2^ test (*p* = 0.00)
Laser trabeculoplasty	15 (18.8)	10 (20.0)	5 (16.7)	Chi^2^ test (*p* = 0.71)
Deep sclerectomy	10 (12.5)	10 (20.0)	0 (0.0)	Chi^2^ test (*p* = 0.01)
Trabeculectomy	3 (3.8)	3 (6.0)	0 (0.0)	Chi^2^ test (*p* = 0.01)

SC = Schlemm’s canal; SD = standard deviation; IOP = intraocular pressure; BCVA = best corrected visual acuity.

**Table 2 jcm-11-00259-t002:** Differences in intraocular pressure and antiglaucoma medications.

		BL	7 Days	15 Days	1 Month	3 Months	6 Months	12 Months
**All eyes**	Eyes, *n*	80	80	79	77	70	68	66
	Mean IOP, mmHg ± SD	22.5 ± 5.3	17.8 ± 7.5	20.5 ± 8.4	17.8 ± 5.5	15.6 ± 4.0	15.3 ± 4.6	15.0 ± 3.6
	Mean IOP reduction from BL, mmHg (%)		4.7 (18%)	1.8 (5.3%)	4.5 (16.6%)	6.5 (26.9%)	6.8 (28.4%)	6.9 (29.0%)
	*p*-Value		(*p* < 0.001)	(*p* = 0.035)	*p* < 0.001	*p* < 0.001	*p* < 0.001	*p* < 0.001
	Mean MEDs, *n* ± SD	3.2 ± 1.0	0.7 ± 1.2	0.9 ± 1.3	1.3 ± 1.4	1.6 ± 1.4	1.7 ± 1.4	1.9 ± 1.4
	Mean MEDs reduction from BL, *n* (*p*-Value)		2.5 (*p* < 0.001)	2.2 (*p* < 0.001)	1.8 (*p* < 0.001)	1.6 (*p* < 0.001)	1.5 (*p* < 0.001)	1.2 (*p* < 0.001)
**Group 1**	Eyes, *n*	50	50	49	47	43	42	40
	Mean IOP, mmHg ± SD	23.0 ± 5.7	17.8 ± 7.9	21.1 ± 9.5	18.6 ± 6.0	16.4 ± 4.4	15.8 ± 5.1	15.6 ± 3.6
	Mean IOP reduction from BL, mmHg (%)		5.2 (20.3%)	1.6 (4.6%)	4.2 (15.1%)	5.9 (23.8%)	6.5 (27.0%)	6.5 (26.8%)
	*p*-Value		*p* < 0.001	*p* = 0.12	*p* < 0.001	*p* < 0.001	*p* < 0.001	*p* < 0.001
	Mean MEDs, *n* ± SD	3.0 ± 1.1	0.7 ± 1.2	0.9 ± 1.2	1.3 ± 1.4	1.6 ± 1.4	1.7 ± 1.4	2.0 ± 1.4
	Mean MEDs reduction from BL, *n* (*p*-Value)		2.3 (*p* < 0.001)	2.1 (*p* < 0.001)	1.7 (*p* < 0.001)	1.4 (*p* < 0.001)	1.2 (*p* < 0.001)	1.0 (*p* < 0.001)
**Group 2**	Eyes, *n*	30	30	30	30	27	26	26
	Mean IOP, mmHg ± SD	21.5 ± 4.7	17.6 ± 7.1	19.4 ± 6.3	16.6 ± 4.5	14.4 ± 2.9	14.5 ± 3.7	14.1 ± 3.3
	Mean IOP reduction from BL, mmHg (%)		3.9 (14.3%)	2.1 (6.2%)	4.9 (19.1%)	7.4 (30.7%)	7.2 (30.3%)	7.6 (32.4%)
	*p*-Value		*p* = 0.007	*p* < 0.069	*p* < 0.001	*p* < 0.001	*p* < 0.001	*p* < 0.001
	Mean MEDs, *n* ± SD	3.4 ± 0.8	0.7 ± 1.1	1.0 ± 1.4	1.3 ± 1.3	1.6 ± 1.4	1.7 ± 1.3	1.9 ± 1.4
	Mean MEDs reduction from BL, *n* (*p*-Value)		2.8 (*p* < 0.001)	2.5 (*p* < 0.001)	2.1 (*p* < 0.001)	1.9 (*p* < 0.001)	1.8 (*p* < 0.001)	1.6 (*p* < 0.001)

IOP = intraocular pressure; SD = standard deviation; BL = baseline; MEDs = medications.

**Table 3 jcm-11-00259-t003:** Success rates overall and in the groups at 12 months.

		Success Rate (%) at 12 Months		
		All Included Eyes	Group 1	Group 2
≤16 mmHg	Complete	14.1	14.0	14.3
	Qualified	58.9	48.0	78.8
≤18 mmHg	Complete	15.4	14.0	17.9
	Qualified	71.8	66.0	82.1
≤21 mmHg	Complete	17.9	18.0	17.9
	Qualified	80.8	76.0	89.3
≤16 mmHg and ≥25% IOP reduction	Complete	10.3	10.0	10.7
	Qualified	43.6	32.0	64.3
≤18 mmHg and ≥25% IOP reduction	Complete	11.5	10.0	14.0
	Qualified	50.0	40.0	67.9

IOP = intraocular pressure.

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
