# Peer review of "360° Ab-Interno Schlemm’s Canal Viscodilation with OMNI Viscosurgical Systems for Open-Angle Glaucoma—Midterm Results"

_jcm, 2022, doi:10.3390/jcm11010259_

Round 1

Reviewer 1 Report

The authors evaluated the results of 360° dilatation of Schlemm's canal using the OMNI system. More options for less invasive glaucoma surgery would be very beneficial for glaucoma specialists.

If possible, please revise the following two points.

1. Please add any figures or intraoperative images to understand this procedure easily.

2. As there are many other options for MIGS, please describe the advantages or disadvantages of this OMNI system.

Author Response

Please see rebuttal letter attached to address comments made by both Reviewers. Thanks

Reviewer 2 Report

This study compared the effect and safety of Viscodilation of SC using OMNI viscosurgical systems alone versus combined with phaco.

The profiles of the demographics are quite different between groups. Statistical analysis is needed in Table 1.

Discussion about the difference between groups in terms of glaucoma type, lens status, and previous surgery and their influence to the results are needed.

Author Response

(The authors gave the same response as above.)

Round 2

Reviewer 2 Report

The authors addressed the issues raised by this reviewer.